# Implementation of Deep Learning Algorithm on a Custom Dataset for Advanced Driver Assistance Systems Applications

Chathura Neelam Jaikishore, Gautam Podaturpet Arunkumar, Ajitesh Jagannathan Srinath, Harikrishnan Vamsi, Kirtaan Srinivasan, Rishabh Karthik Ramesh, Kathirvelan Jayaraman * and Prakash Ramachandran

School of Electronics Engineering, Vellore Institute of Technology, Vellore 632014, India
* Correspondence: j.kathirvelan@vit.ac.in

**Abstract:** Road hazards such as jaywalking pedestrians, stray animals, unmarked speed bumps, vehicles, and road damage can pose a significant threat in poor visibility conditions. Vehicles are fitted with safety technologies like advanced driver assistance systems (ADAS) and AW (automatic warning) systems to tackle these issues. However, these safety systems are complex and expensive, and these proprietary systems are exclusive to high-end models. The majority of the existing vehicles on the road lacks these systems. The YOLO model (You Only Look Once Architecture) was chosen owing to its lightweight architecture and low inference latency. Since YOLO is an open-source architecture, it can enhance interoperability and feasibility of aftermarket/retrofit ADAS devices, which helps in reducing road fatalities. An ADAS which implements a YOLO-based object detection algorithm to detect and mark obstacles (pedestrians, vehicles, animals, speed breakers, and road damage) using a visual bounding box was proposed. The performance of YOLOv3 and YOLOv5 has been evaluated on the Traffic in the Tamil Nadu Roads dataset. The YOLOv3 model has performed exceptionally well with an F1-Score of 76.3% and an mAP (mean average precision) of 0.755, whereas the YOLOv5 has achieved an F1-Score of 73.7% and an mAP of 0.7263.

**Keywords:** ADAS; YOLOv3; YOLOv5; deep learning; object detection; road safety

## 1. Introduction

The number of road accident fatalities in India has been alarming. According to the accidental deaths and suicides in India reported by the National Crime Records Bureau of India in 2019, 154,732 fatalities and 439,262 injuries were caused due to road accidents. An alarming 38% of the road fatalities involved two-wheelers, while 14.6% and 13.7% involved trucks and cars, respectively. Furthermore, 7.7% of the total fatalities were pedestrians [1]. Advanced driver assistance systems can significantly reduce road collisions and related fatalities by up to 47%. A study based on data from Polish road crashes in 2015 indicated that wide deployment of ADAS systems such as advanced emergency braking systems (AEBS), adaptive cruise control (ACC), and lane departure warning (LDW) systems has the potential to reduce road crashes by 47% when deployed in combination. AEBS alone has the potential to reduce the crash rate by 33%. Meanwhile, ACC alone can reduce the crash rate by 27%, and LDW alone by 4% when deployed widely [2].

## 2. Literature Review

Many researchers and authors conducted research and presented their results. For instance, Khan et al., in 2019, [3] conducted research on the personal and societal benefits of ADAS systems such as blind-spot monitoring, lane departure warning, and forward-collision warning in light-duty vehicles sold in the United States in 2015. Their research estimated that the technologies above can collectively prevent up to 1.6 million crashes every year, including 7200 fatal crashes. Thus, ADAS adoption will lead to positive personal and societal benefits. A study [4] evaluated the field effectiveness of ADAS technologies

such as frontal collision warning with autonomous emergency braking, lane departure warning, and blind-spot detection deployed in BMW cars sold in the United States between 2014 and 2017 in preventing moderate-to-severe crashes. Their research shows that in the 2014 model year, vehicles equipped with ADAS were 13% less likely to crash than their non-ADAS versions. Similarly, in the 2017 model year, vehicles were 34% less likely to crash when all the systems above were deployed together. Enhancing the hazard warning systems available to drivers and automated systems such as AEBS (advanced emergency braking systems), pedestrian warning, and blind-spot warning systems with visual and auditory warning can help improve the driver's behavior on the road. Thompson in 2018 [5] evaluated driver behavior in a trial of retrofitting the CAT (collision avoidance technology) ADAS system in government fleet vehicles, which provided LDW and FCW (forward collision warning) in the form of audio and visual warnings to drivers. The analysis from the driver feedback survey and other vehicle parameters from the CAT system indicated that the system effectively improved drivers' behavior on-road and helped reduce crashes. Nevertheless, the drivers felt that such a system was distracting and would not use it themselves but agreed that it would improve overall road safety. The authors suggested that optimism bias might be a reason for such feedback from the drivers and concluded that appropriate training about the CAT system will be useful in increasing the acceptance and perception of such a system among drivers. This was reiterated by Oviedo-Trespalacios et al. [6], that there is a need for proper driver education on ADAS systems to improve driver behavior and effective system use in improving overall road safety and preventing collisions. Lack of standardization and appropriate visual guides leads to an incomplete understanding of the system's behavior and limitations.

Camera sensors that provide computer vision-based object detection capability using deep learning-based algorithms are a vital part of ADAS across all levels of automation. From a simple level 0 camera-based parking assist system to a comprehensive level 5 fully autonomous vehicle, the computer vision-based object detection system acts as the eyes in the environment to the ADAS based on which crucial warnings, decisions, and automation tasks are accomplished [7]. A study by Benjdira et al. [8] compared the performance of two state-of-the-art convolutional neural networks (CNN) algorithms, namely, faster R-CNN (region-based CNN) and YOLOv3, for detecting cars from aerial images. Both models were trained and tested on a large car dataset taken from UAVs. These authors demonstrated in this paper that YOLOv3 outperforms faster R-CNN in terms of sensitivity and processing time, despite being similar in terms of precision. Al-qaness et al. [9] developed an intelligent vehicle tracking system based on video surveillance. In order to track vehicles, this system integrates neural networks with image-based tracking, and You Only Look Once (YOLOv3). Different datasets were used to train this system. Babayan et al. [10] presented a comparison of different neural network architectures for object detection and recognition. Pedestrians and vehicles were the subjects of their study. A study by Byeon and Kwak [11] presented a performance comparison of pedestrian detection using faster R-CNN and aggregated channel features (ACF). CNN's feature vectors are extracted from each regional image by R-CNN independently looking for candidates for detectable objects. Support-vector machines (SVMs) are then used to classify the images. The ACF algorithm combines several channels from one image and obtains a low-resolution channel by integrating them into a smoothed one. Pixels turn into features, and these features turn into feature vectors. By using decision trees and boost, pedestrians and background are separated. Rahman et al. [12] used the Dhaka Traffic Detection Challenge Dataset (DhakaAI) to train and evaluate the performance of the YOLOv5 architecture-based model pretrained on the COCO dataset with 17 classes out of 80 available for vehicles in the DhakaAI dataset to detect vehicular objects in densely crowded images for real-time application. They achieved a maximum (mean average precision) mAP (@ 0.5) of 0.485 on an NMS (nonmaximum suppression)-based ensemble of four models trained using the YOLOv5 model pretrained using the COCO dataset and then evaluated on the DhakaAI dataset.

## 3. Technical Overview

### 3.1. The Evolution of the ADAS System

Advanced driver assistance systems (ADAS) is a driver assistance technology designed to enhance a driver's ease of driving and enhance road safety. The difference between advanced driver assistance systems and traditional driver assistance systems lies in the level of automation provided and the sensor set used. Traditional driver assistance systems such as ABS (antilocking braking systems) measure the onboard vehicle parameters and actuate the control mechanism to improve driver safety and comfort. In this case of the antilock braking system, the wheel speed sensor measures the wheel speed and continuously actuates a brake valve to prevent locking of the wheels during braking. In the case of advanced driver assistance systems such as frontal collision avoidance systems, the surrounding environment is sensed using sensors to warn about driver hazards and perform automatic control or maneuvers to prevent mishaps. This enhances the safety and comfort of drivers. In this case, computer vision-based object detection identifies the hazard via a camera. A radar-based sensor is used to localize the hazard, i.e., estimate the distance of the hazard to perform corrective action (audio, visual, or haptic feedback warning). There are five levels of vehicle driving automation specified by SAE (Society of Automotive Engineers), from level 1 to level 5.

- Level 0—No Automation: The system performs no automation (control task) but only provides information on onboard or external parameters measured by the vehicle to the driver for his information or warning. This system includes the surround view parking assistance system which uses traditional ultrasonic sensors combined with a camera to provide the driver with a video stream where vehicle lines, curbs, walls, and obstructions are superimposed and highlighted on the video stream to assist the driver. The traffic sign recognition system assists drivers by identifying and providing information on current road rules posted on traffic signs. The system uses the camera video feed from the windshield as input. With the help of computer vision-based text recognition and object detection, deep learning algorithms recognize the information on the signs and provide information and warnings to the driver. A few common applications belonging to Level 0 are:
    - Lane Departure Warning: This provides an audio and/or visual warning to the driver if the vehicle is accidentally deviating from the current lane. It is implemented with the help of a forward-facing camera.
    - Night Vision: helps the driver with better visibility of the road in low light and dark conditions by using a camera feed and an IR illuminator.
    - Blind Spot Detection: provides a visual and optionally auditory warning about obstacles present in the driver's rear-view blind spot using two short-range radars, which are fixed in the rear corners of the vehicle.
    - Forward Collision Warning: makes use of a combination of mid-range RADAR as well as a front-facing camera to provide the driver with audio and visual warnings about an obstacle ahead that may collide with the vehicle.
- Level 1—Driver Assistance: these systems assist the driver in performing the driver's tasks by controlling a single specific driving function, but they leave the driver in complete control of the vehicle and require them to be alert at all times.
- Level 2—Partial Automation: it is similar to level 1 systems but provides additional automation in the form of combined automated intervention of various level 1 systems but requires the driver to be alert all the time.
- Level 3—Conditional Automation: these systems do not require the driver to be alert all the time, but the driver should be ready to take control of the vehicle at any time if the system determines that it cannot handle the situation and alerts the driver.
- Level 4—High Automation: This system is an extension of Level 3 automation, where the system can handle unknown situations based on its decision-making intelligence

without the driver's intervention. It may still require a limited amount of driver intervention in certain conditions.

- Level 5—Full Automation: This level of automation is a scenario where the driver's intervention is never required for all scenarios. The system is fail-safe and fail-operational, i.e., it has the ability to handle unknown scenarios without the driver's intervention. The driver may still control the vehicle manually voluntarily (Galvani 2019).

### 3.2. Technical Overview of Artificial Intelligence

A classical neural network (NN) can recognize patterns and classify different types of information. Layers serve as filters, thereby increasing the likelihood of finding the best output. Deep learning and convolutional neural networks are essential tools for detecting objects. The two main categories of deep learning algorithms are single-stage and two-stage classifiers. A two-stage algorithm generates regions containing objects. A neural network then classifies these regions into objects. Consequently, they are generally more accurate than single-stage classifiers. However, their inference speed is slower because of the multiple stages involved in detecting anomalies. Alternatively, in single-stage detectors, the region proposal step is removed, and object classification and localization coincide. As a result, single-stage classifiers are faster than two-stage classifiers. The R-CNN method first generates potential bounding boxes in an image and then runs a classifier on them. Following classification, postprocessing refines the bounding boxes, removes duplicates, and re-scores them with other objects in the scene. As each component of this complex pipeline must be trained separately, the entire process becomes slow and difficult to optimize [13]. However, the YOLO model reframed the object detection task as a single regression problem, with image pixels directly translated into bounding box coordinates and class probabilities. In this method, objects are only seen once in an image to predict where they are. This method is simple but effective. With a single convolutional network, multiple bounding boxes can be predicted, along with their class probabilities. YOLO optimizes performance directly by training on full images, improving speed and accuracy [14]. There are different versions of YOLO, and the major incremental improvements are YOLOv1, YOLOv2, YOLOv3, YOLOv4, and YOLOv5, tiny-YOLOv3, and tiny-YOLOv4. The main focus of this paper was to compare the performance of YOLOv3 and YOLOv5 on an ADAS application.

#### 3.2.1. YOLOv3

Darknet53, which acts as the backbone for YOLOv3, combines a convolution layer and a deep neural network. This convolution layer first extracts features from the image and feeds them to the feature pyramid network (FPN) to fuse them. The FPN is used as the neck. The neck plays an important role in extracting feature maps from different stages, consisting of several bottom-up and top-down paths, and the head contains YOLO maps. As part of a one-stage detector, the head calculates the final prediction based on bounding box coordinates: width, height, class label, and probability of each class. Finally, the result is derived from the YOLO layer [15].

#### 3.2.2. YOLOv5

The next incremental version, YOLOv5, differs from its predecessors by using PyTorch instead of the Darknet framework. The backbone is CSPDarknet53. CSPDarknet53 extracts features from the image. Repetitive gradient information in large backbones is eliminated, and gradient change is incorporated into feature mapping. These changes reduce inference speed, improve accuracy, and reduce the model size by shrinking the parameters. To boost information flow, PANet is used as the neck. The PANet improves localization in lower layers, which increases the object's localization accuracy. YOLOv5 also uses the same head as YOLOv3, generating three different output vectors and achieving multiscale prediction. The results are generated using the final YOLO layer. YOLOv3 and YOLOv5 architectures

differ mainly by using the Darknet53 backbone. CSPdarknet53 is used as the backbone for YOLOv5. YOLOv5 introduced the Focus layer. In YOLOv5, the Focus layer replaces YOLOv3's first three layers. The Focus layer reduces CUDA memory usage, reduces layers, and increases forward propagation and also back-propagation [16].

## 4. Materials and Methods

The overview of the proposed architecture is illustrated in Figure 1 and every step is explained in detail in this section.

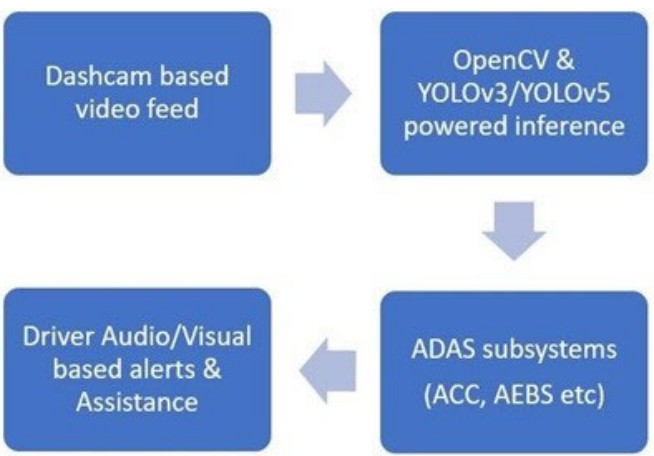

**Figure 1.** The overview of the proposed methodology.

### 4.1. Dataset Generation

A Raspberry Pi 8 MP camera interfaced with a Raspberry Pi controller, Figure 2 setup, was installed on the vehicle's dashboard. The streamed output is in the form of 640 × 480 pixel mpeg frames at a rate of 24 frames per second. The dataset was generated by driving a car through various road conditions and generating a video of nighttime and daytime driving in the city and highway. The video generated using the Raspberry Pi camera is sampled at one frame per second to generate a dataset with 5945 images. This dataset was uploaded to Kaggle and named as Traffic in Tamil Nadu Roads, Figure 3. The images were manually annotated with seven different classes (4WD, 2WD, Pedestrians, Stray Animals, Speed Bumps, Road Damage, and Barricades) using make sense.ai [17] to suit the Yolo training and labelling format for each image. The generated set of annotations for the dataset was downloaded as a zip folder containing individual text files for each image. Each text file contained a row of five values (the label encoded value of the class name, the X and Y coordinates of the center of the bounding box, and the bounding box's width and height) for each object in the image.

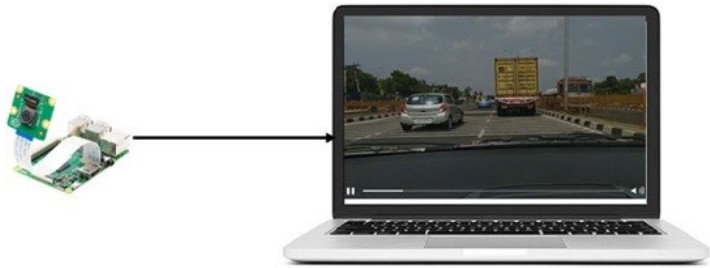

**Figure 2.** The Raspberry Pi camera setup.

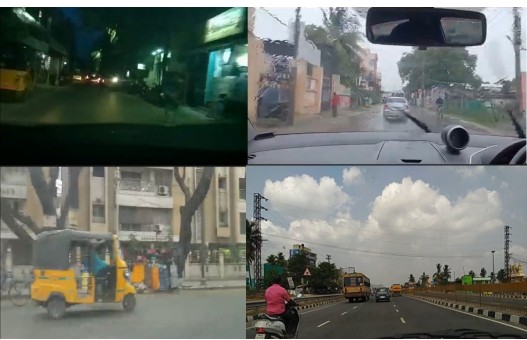

**Figure 3.** The sample dataset.

### 4.2. Model Training

This dataset was split into training and testing data (80:20), allocating 4482 images to the training set and 1463 images to the test set. The images were resized to 416 × 416 pixels. The batch size is chosen to be 16 and runs for 20 epochs. The learning rate is chosen as 0.0001 and SGD is the chosen optimizer. The dataset was trained using pretrained weights (yolo5x.pt for YOLOv5 and YOLOv3.pt for YOLOv3) utilizing transfer learning techniques.

The architectural views of YOLOv3 and YOLOv5 are illustrated in Figures 4 and 5. The pretrained weights are cloned from the ultralytics GitHub repository. The same dataset was also trained without pretrained weights where the weights were initialized randomly [18].

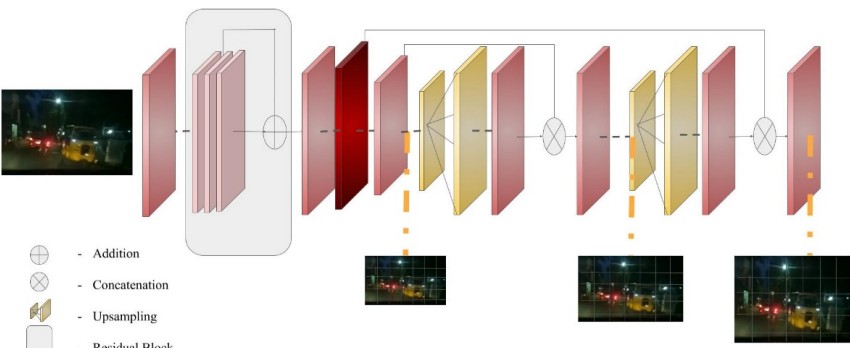

**Figure 4.** The YOLOv3 architecture.

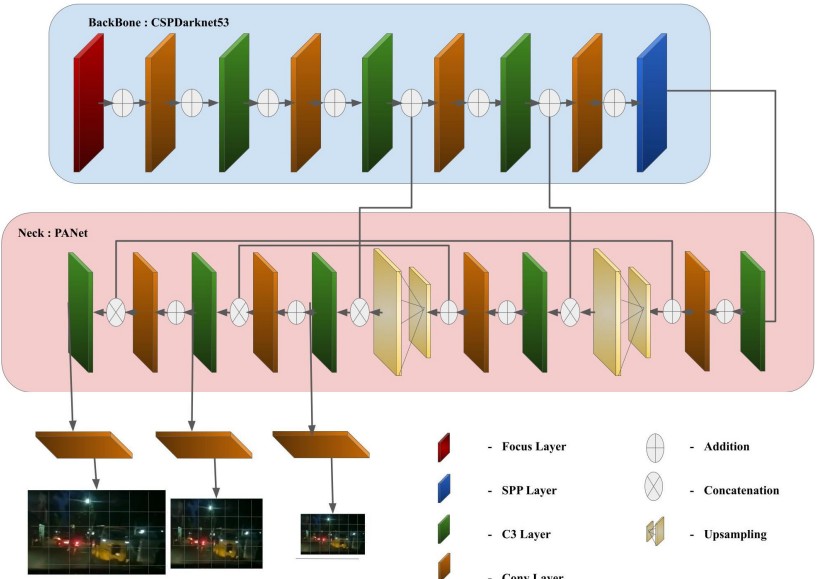

**Figure 5.** The YOLOv5 architecture.

### 4.3. Model Testing

After training the models, the best and the last weights of the neural network layers are stored in the weights folder. We utilized the best weights with the best metrics for the final detection of the objects with a confidence of 40% or more. Each model is applied to 3 videos (rainy, daylight, and night). The sample of output from the YOLO models is illustrated in Figures 6 and 7. The detection of four-wheelers, two-wheelers, and other classes can be seen in Figures 6 and 7 with a bounding box around the vehicles along with their respective labels.

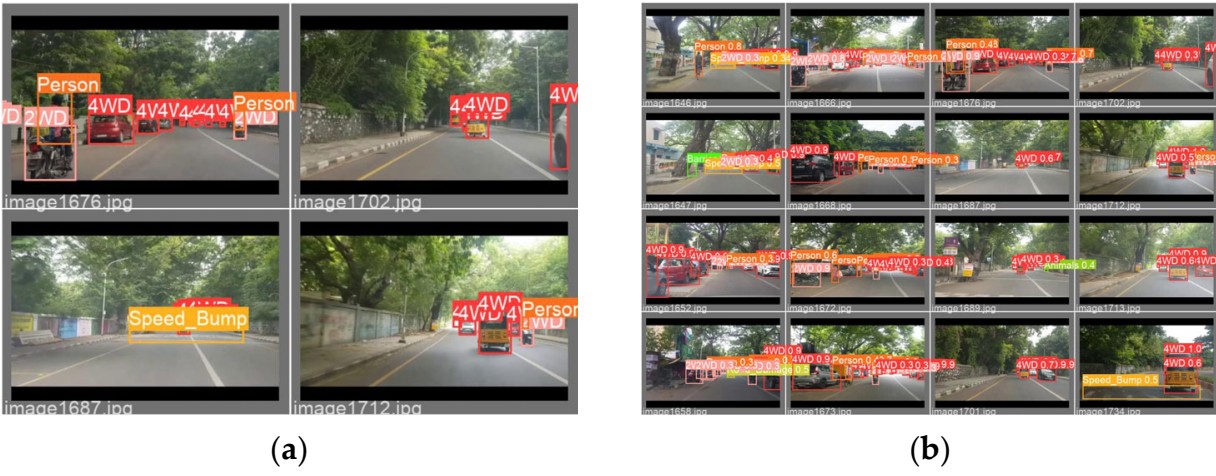

(**a**)                                                         (**b**)

**Figure 6.** The test results of (**a**) YOLOv3 model without pretrained weights and (**b**) YOLOv3 model with pretrained weights.

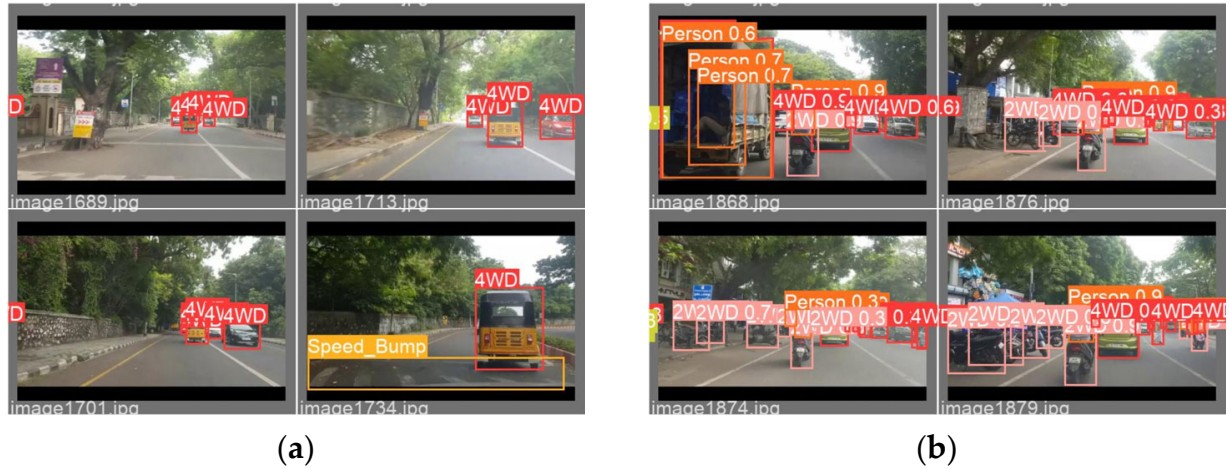

(**a**)                                                         (**b**)

**Figure 7.** The test results of (**a**) YOLOv5 model without pretrained weights and (**b**) YOLOv5 model with pretrained weights.

Both the models were tested on a 1:24-long video, which tallied to about 2500 frames. The overview of training the models on this dataset is presented in Figure 8. A comparative study is presented in further sections.

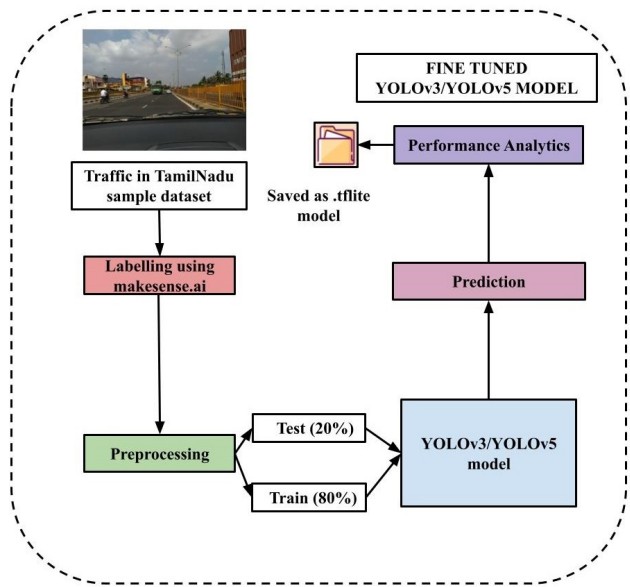

**Figure 8.** The flow diagram of the YOLO model.

### 4.4. Mobile Application

A mobile application was developed using Android Studio. The YOLO models were compatible with this application since the model had virtually no latency. The application was developed using Java. Classes were developed to access the camera and also to detect activity on the road to draw the bounding box around the detected objects. The models are saved in tflite (Tensor Flowlite) format and incorporated into the mobile application. The app works well with optimal inference time on 640 × 480 frames. The app requests permission to access the camera when it is installed for the first time. Figure 9 presents screenshots of the developed prototype. Figure 9a is the initial, blank UI with just the recording button. The application uses the phone's in-built camera to capture the road. In response to being tapped, a live feed of the road along with the prediction of the obstacles is relayed. The image is passed to the YOLO model to determine the bounding box and the class of the obstacle. We can observe in Figure 9b that the four-wheeler vehicle is detected correctly. Because smartphones are common, a simple user interface in an Android application that functions without latency makes this application very user-friendly.

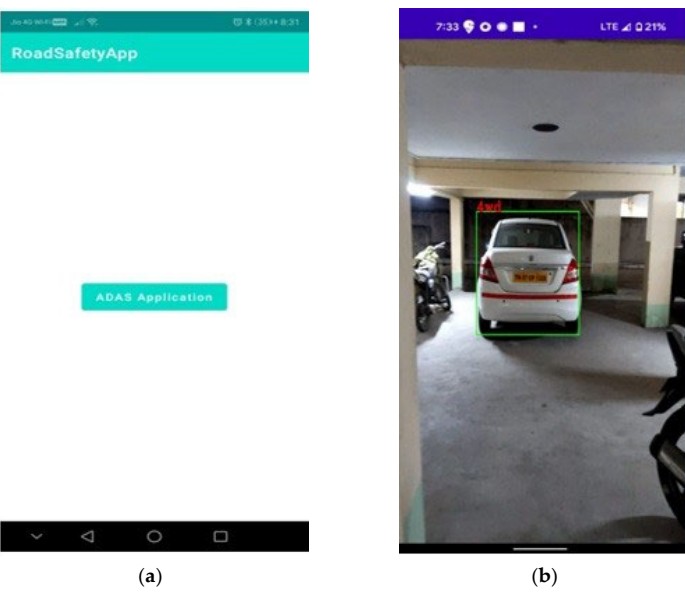

(**a**)                    (**b**)

**Figure 9.** (**a**) The initial UI blank screen. (**b**) The road safety app running on the mobile phone.

## 5. Performance Comparison

The accuracy of the model is evaluated using the F1 score, mean average precision (mAP), and the Precision x Recall curve. It is essential to examine the outcomes of both the classification (confusion matrix) and localization (using the IoU of bounding boxes in the image) metrics of the model.

**Intersection over Union (IoU)** is employed while detecting objects. IoU computes an intersection over the union of the two bounding boxes: the bounding box for the ground truth and the predicted bounding box. When the IoU is 1, it implies that the bounding boxes for predicted and ground truth overlap totally. A bounding box is rendered around an object only if the IoU is greater than the set threshold value. In this application, a threshold of IoU is set to be 0.4. As a result, when Iou > 0.4, the detection is considered true positive (TP), whereas if IoU < 0.4, it is classified as false positive (FP). If the model fails to detect the object in the image despite having ground truth, it is classified as false negative (FN). An image's true negative (TN) is any portion where no object can be predicted. It does not help detect objects. Therefore, TN is ignored in Equation (1).

$$IoU = \frac{Area\ of\ Overlap}{(Area\ of\ Union)} \quad (1)$$

**Precision**, according to the definition, is the proportion of positive samples correctly classified to all positive samples (whether correctly or incorrectly classified). It measures how accurate a model is at classifying a sample as positive, as mentioned in Equation (2). The comparison among the various precision curves is illustrated in Figures 10 and 11.

$$Precision = \frac{TP \times 100}{(TP + FP)} \quad (2)$$

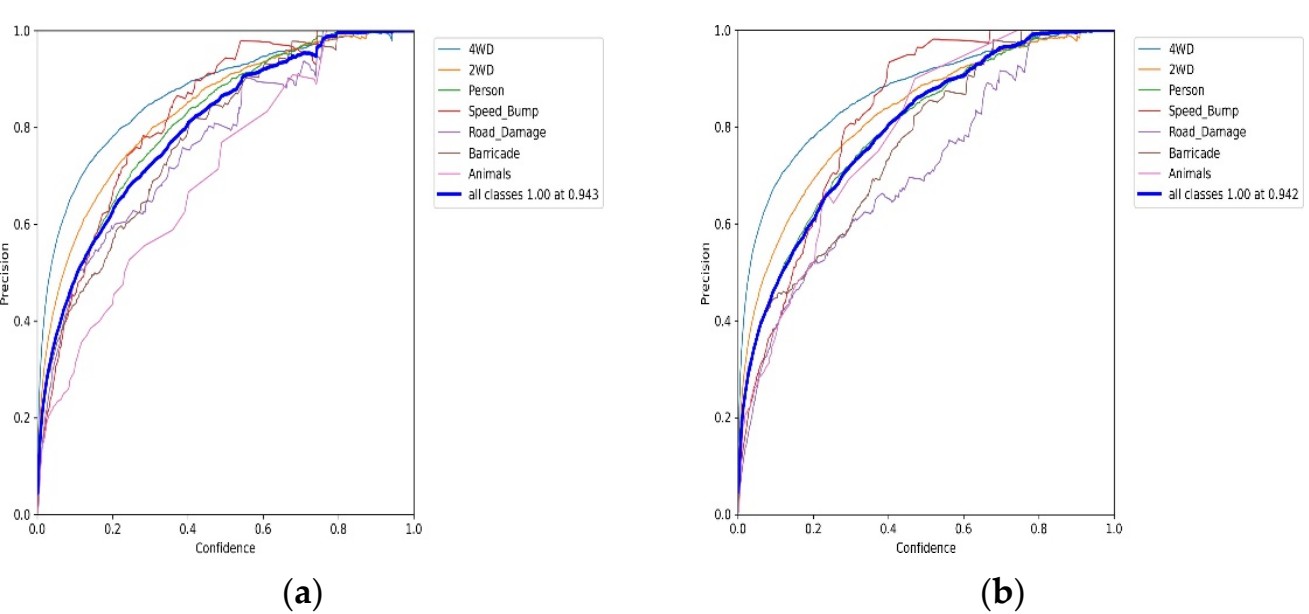

**(a)** **(b)**

**Figure 10.** Precision curve of (**a**) YOLOv3 model without the pretrained weights, (**b**) YOLOv3 model with the pretrained weights.

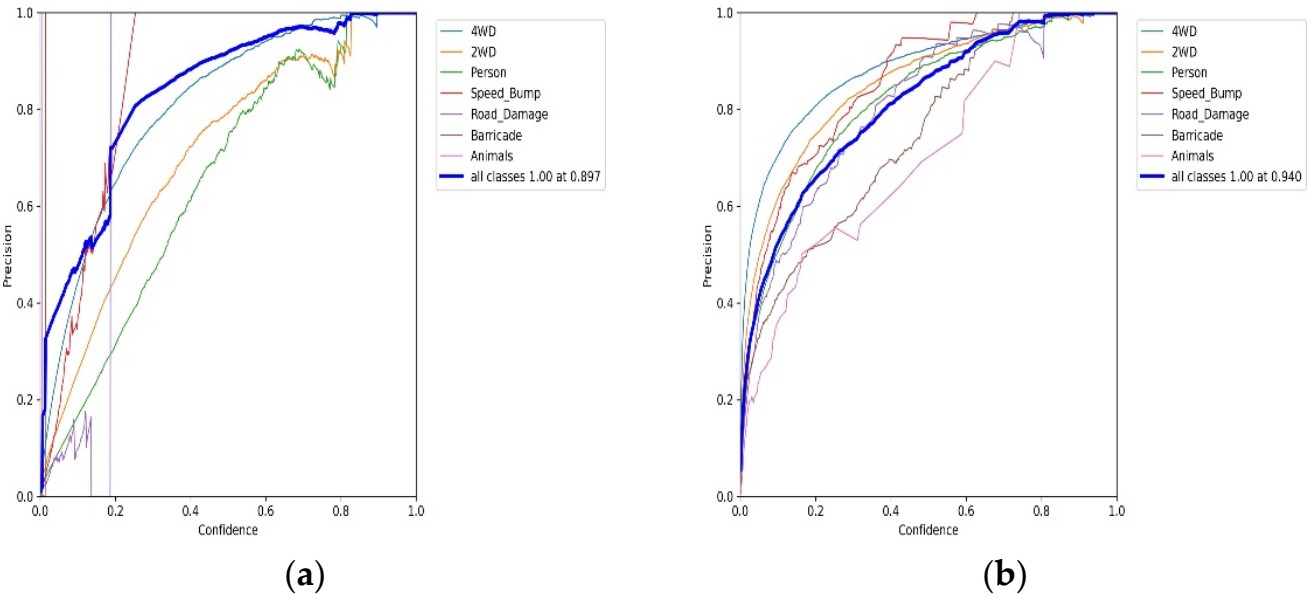

**(a)** **(b)**

**Figure 11.** Precision curve of (**a**) YOLOv5 model without the pretrained weights; (**b**) YOLOv5 model with the pretrained weights.

**Recall** is calculated by taking the ratio of the number of positive samples correctly classified as positive to the total number of positive samples. This recall indicates how well the model detects positive samples and is quoted in Equation (3). The comparison of recall curves with respect to the ADAS application is pictured in Figures 12 and 13.

$$Recall(Sensitivity) = \frac{TP \times 100}{(TP + FN)} \tag{3}$$

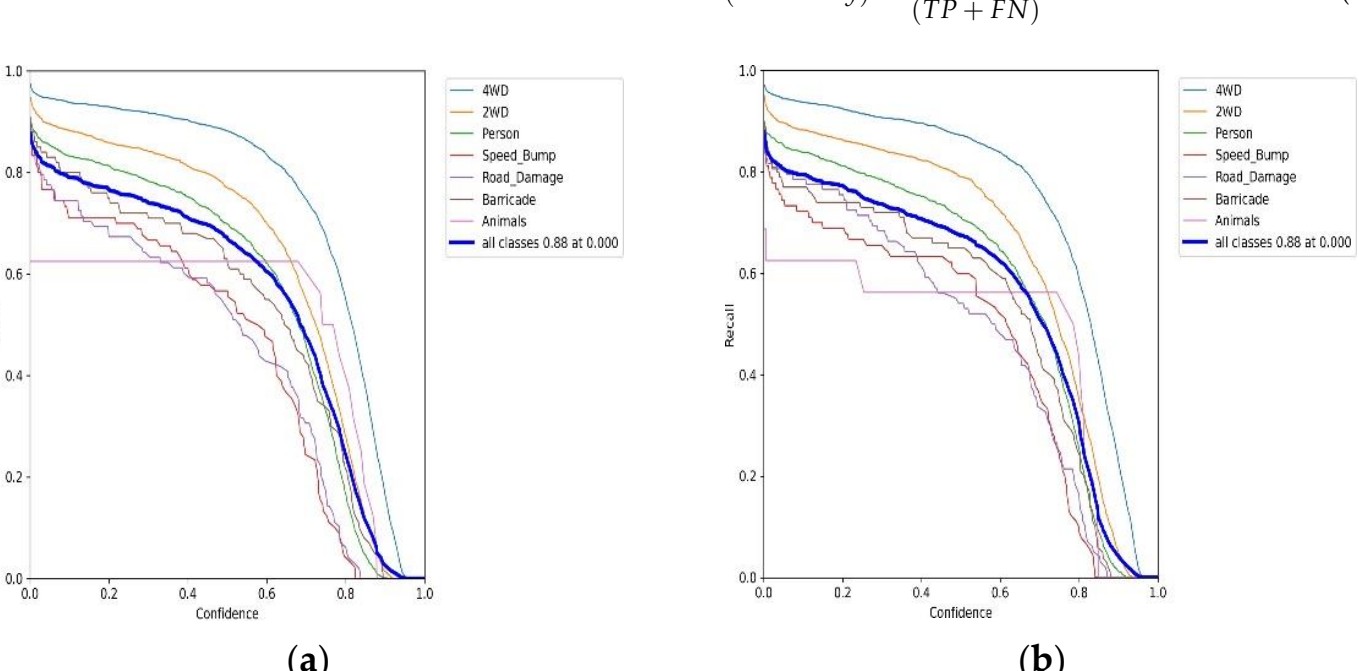

**(a)** **(b)**

**Figure 12.** Recall curve of (**a**) YOLOv3 model without the pretrained weights; (**b**) YOLOv3 model with the pretrained weights.

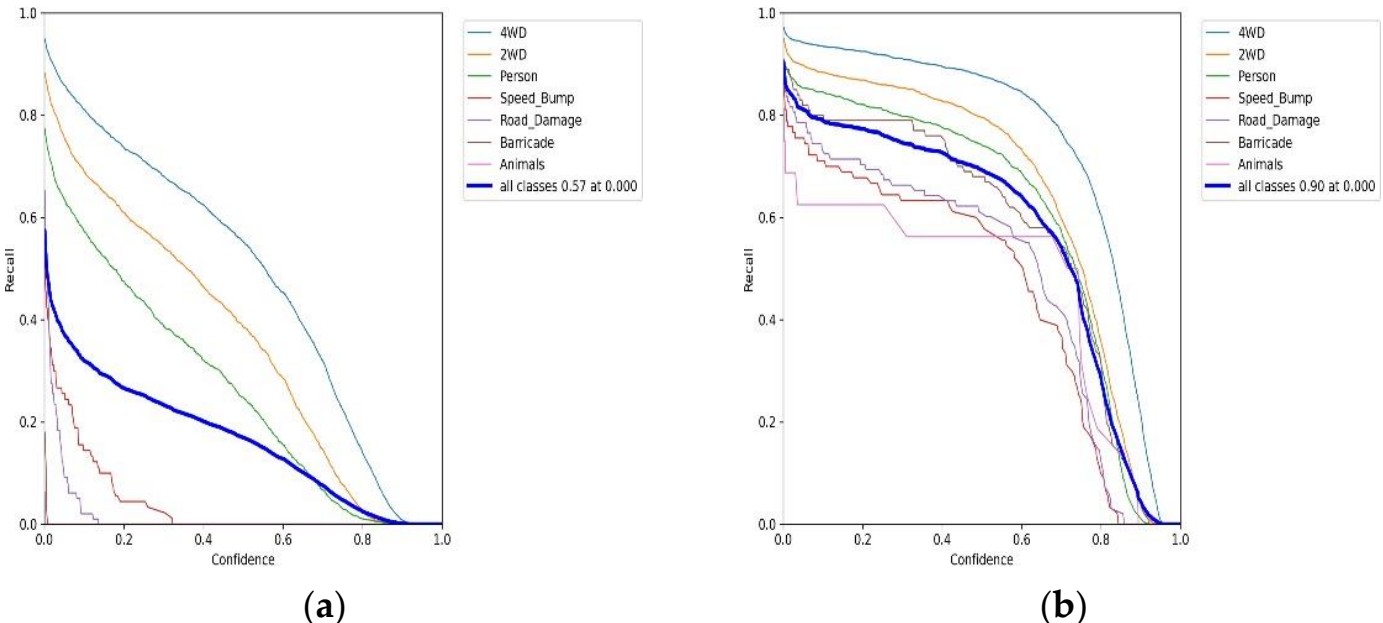

**Figure 13.** Recall curve of (**a**) YOLOv5 model without the pretrained weights; (**b**) YOLOv5 model with the pretrained weights.

**F1 score** is calculated from the precision and recall of each observation. A score of 1 indicates maximum accuracy, which is depicted using Equation (4). The comparison among various models for the ADAS application is presented in Figures 14 and 15.

$$F1\ Score = \frac{2 \times (Recall \times Precision)}{(Recall + Precision)} \tag{4}$$

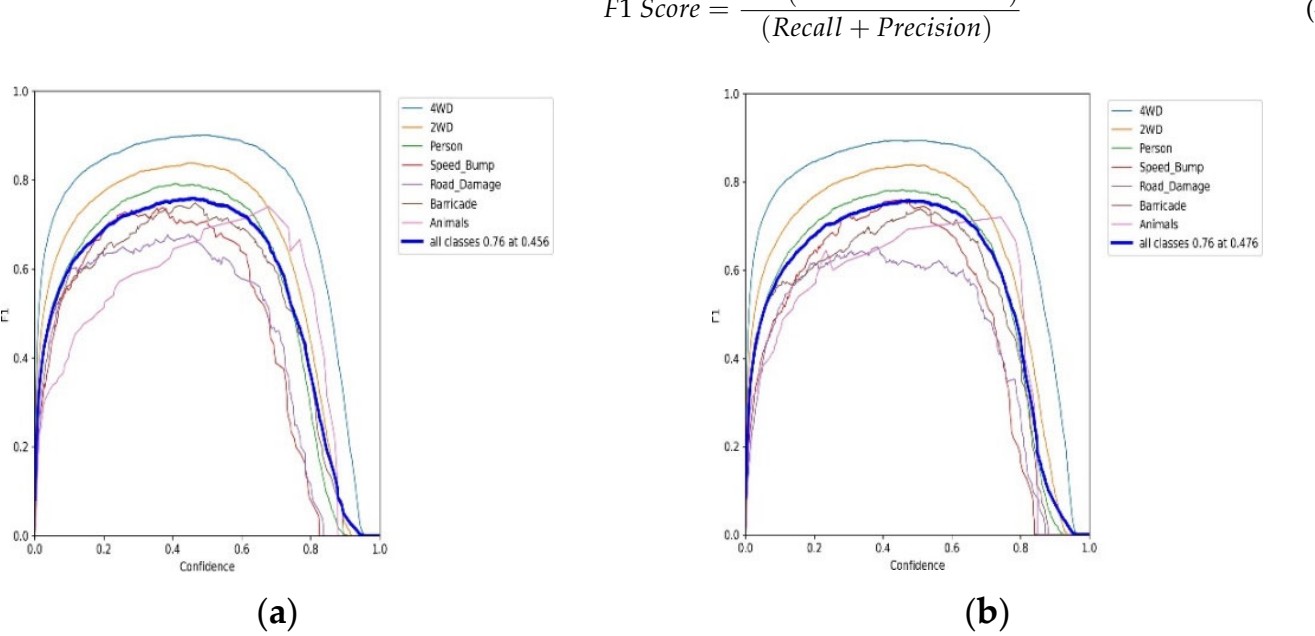

**Figure 14.** F1-Score curve of (**a**) YOLOv3 model without the pretrained weights; (**b**) YOLOv3 model with the pretrained weights.

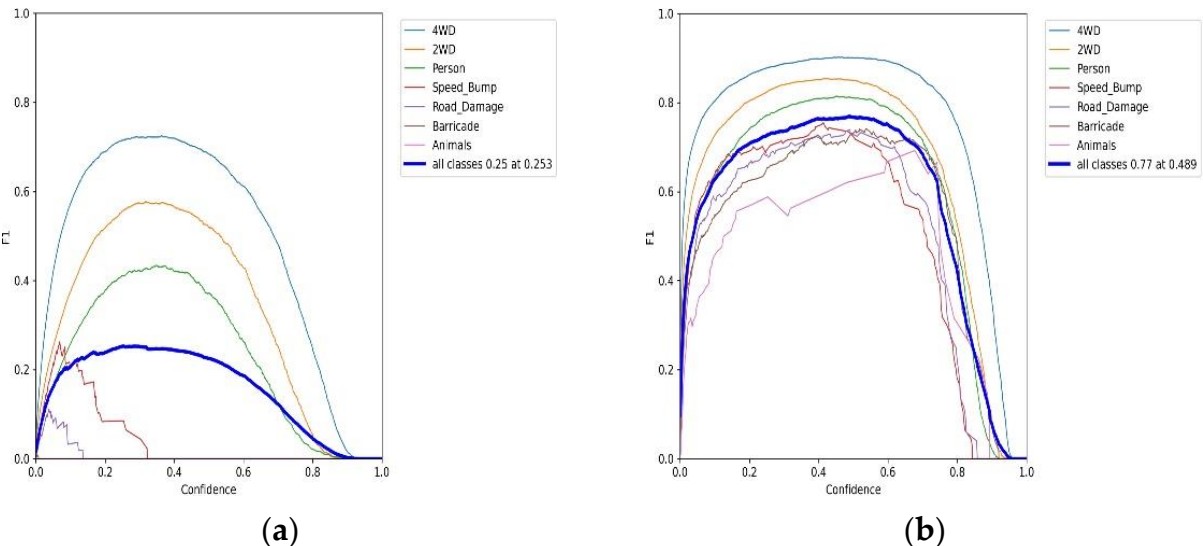

**Figure 15.** F1-Score urve of (**a**) YOLOv5 model without the pretrained weights; (**b**) YOLOv5 model with the pretrained weights.

The *mAP* **score** is determined by comparing the ground truth bounding box to the detected box. The higher the score, the better the model's detection. This is mathematically illustrated in Equation (5).

$$mAP = \frac{\sum_{q=1}^{Q} Q \rightarrow AP(q)}{Q} \tag{5}$$

**PR-Curve** Precision decreases as recall increases. When the number of positive samples increases (high recall), the accuracy of identifying the sample decreases (low precision). It is possible to determine where precision and recall are high by looking at the precision–recall curve. The main factor that contributes to picking the model with the right trade-off between precision and recall is pictured in Figures 16 and 17. Here, TP denotes the number of true positive cases, TN is the number of true negative cases, FP indicates the number of false positive cases, FN is the number of false negative cases, and Q denotes the number of queries [19].

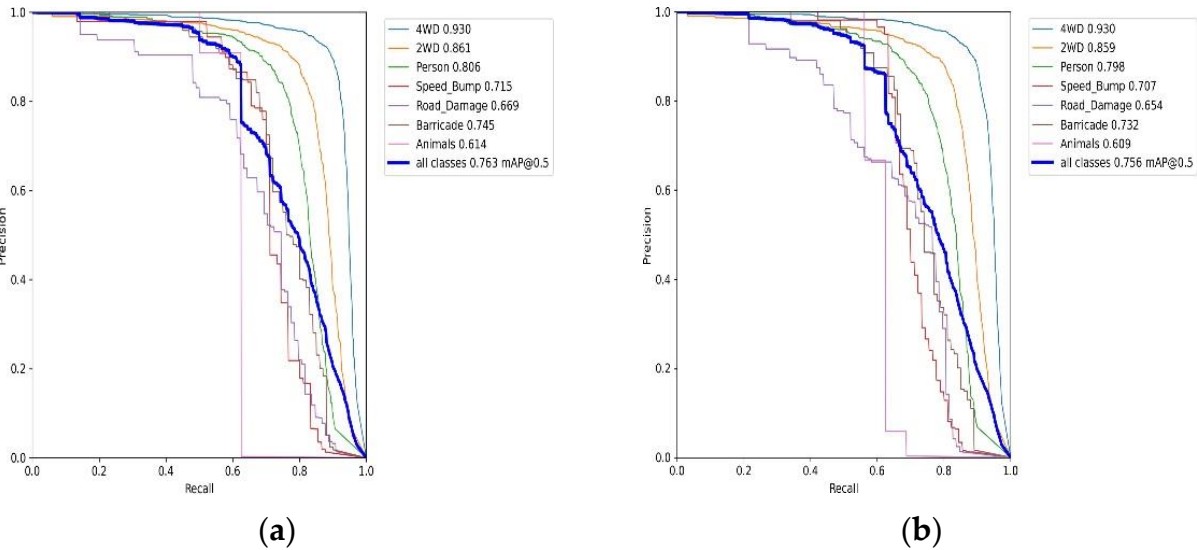

**Figure 16.** Precision–recall curve of (**a**) YOLOv3 model without the pretrained weights; (**b**) YOLOv3 model with the pretrained weights.

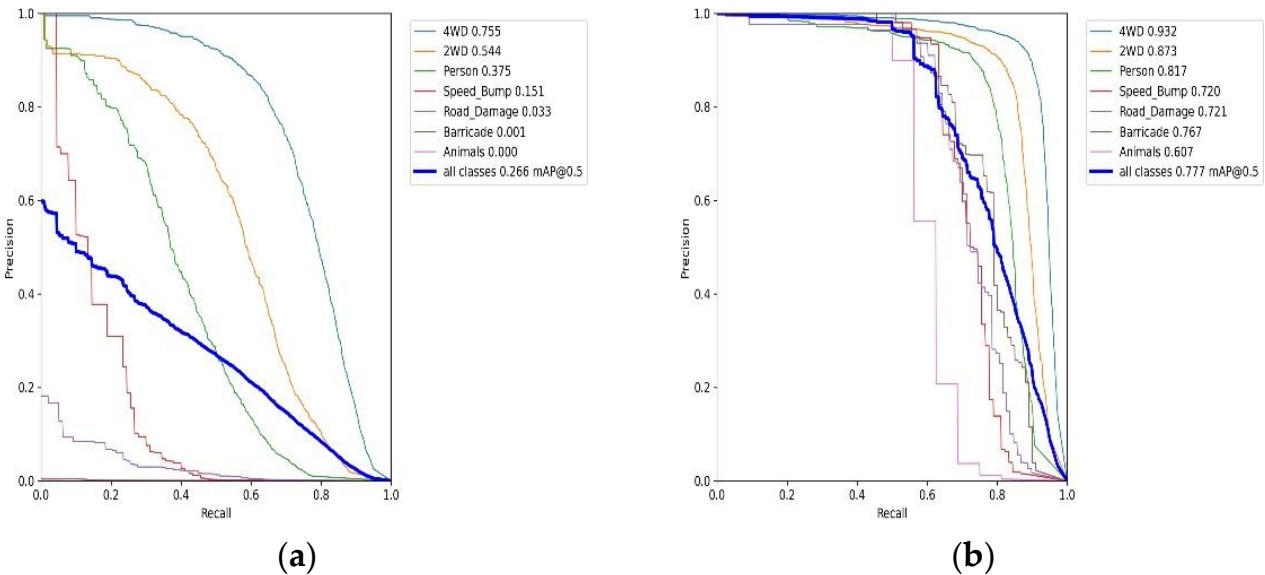

**Figure 17.** Precision–recall curve of (**a**) YOLOv5 model without the pretrained weights; (**b**) YOLOv5 model with the pretrained weights.

From the above Table 1, it is evident that, in general, YOLOv3 models have performed better than YOLOv5 in terms of F1 Score and mAP. The performance difference between the YOLOv3 model with and without pretrained weights is negligible. YOLOv3 with pretrained weights performed slightly better than the YOLOv3 model without pretrained weights in terms of F1 Score (0.76326); whereas the latter performed better in terms of mAP (0.7626). In the case of YOLOv5, the model without pretrained weights performed poorly in terms of both F1 score and mAP compared to the model with pretrained weights [F1-Score (0.7378) and mAP (0.26588)]. The performance overview of all the models is presented in Figures 18–21. Even with an NMS (nonmaximum suppression) ensemble of four models trained using YOLOv5 of the DhakaAI dataset, the YOLOv5 managed to achieve an mAP (@ 0.5) of only 0.458 as mentioned in Section 1. The performance of a single-fold trained YOLOv5 model on the DhakaAI dataset was low compared to our model.

**Table 1.** Comparison of the proposed work with the existing works.

| Sno Citation | F1-Score | mAP (>0.5) |
|---|---|---|
| 1 Benjdira et al., 2019 | 99.94% | N/A |
| 2 Al-qaness et al., 2021 | N/A | 88% |
| 3 Nepal and Eslamiat 2022 | 65.5% | 63.3% |
| 4 Babayan et al., 2019 | N/A | 88.3% |
| 5 Byeon and Kwak 2017 | 24.24% | N/A |
| 6 Rahman et al., 2022 | N/A | 45.8% |
| **7 Proposed Work** | 76.32% | 75.55% |

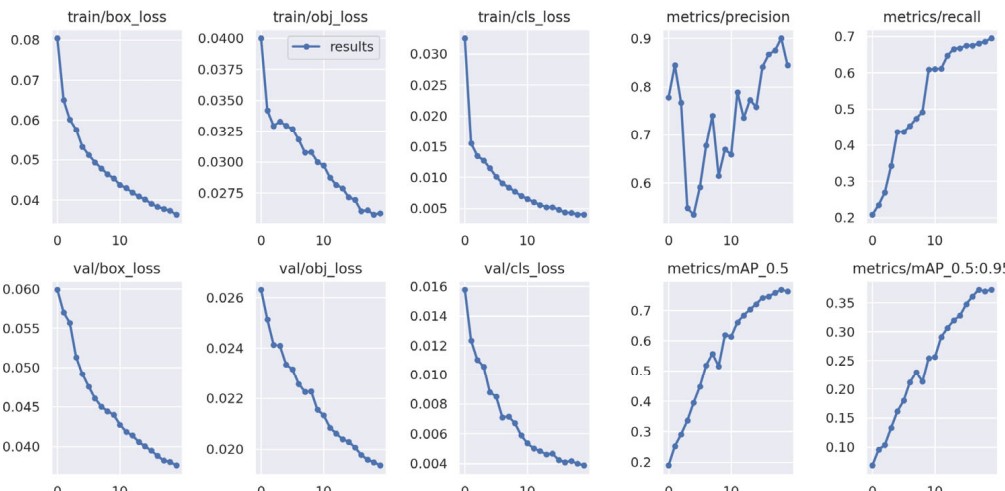

**Figure 18.** Comprehensive results of YOLOv3 model without the pretrained weights.

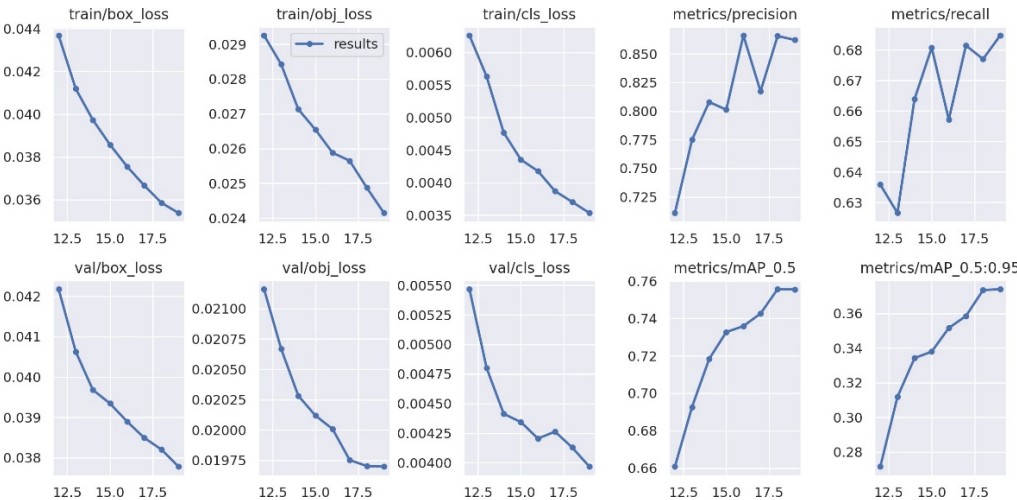

**Figure 19.** Comprehensive results of YOLOv3 model with the pretrained weights.

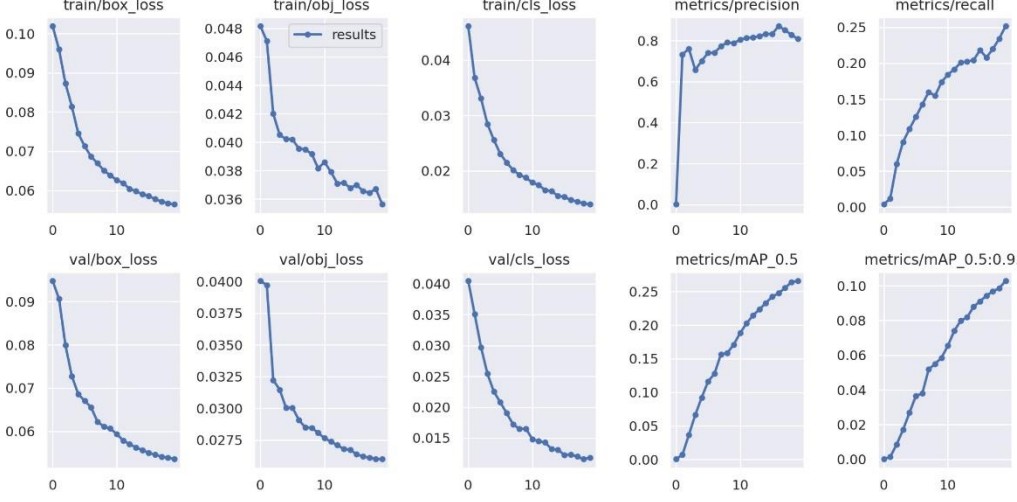

**Figure 20.** Comprehensive results of YOLOv5 model without the pretrained weights.

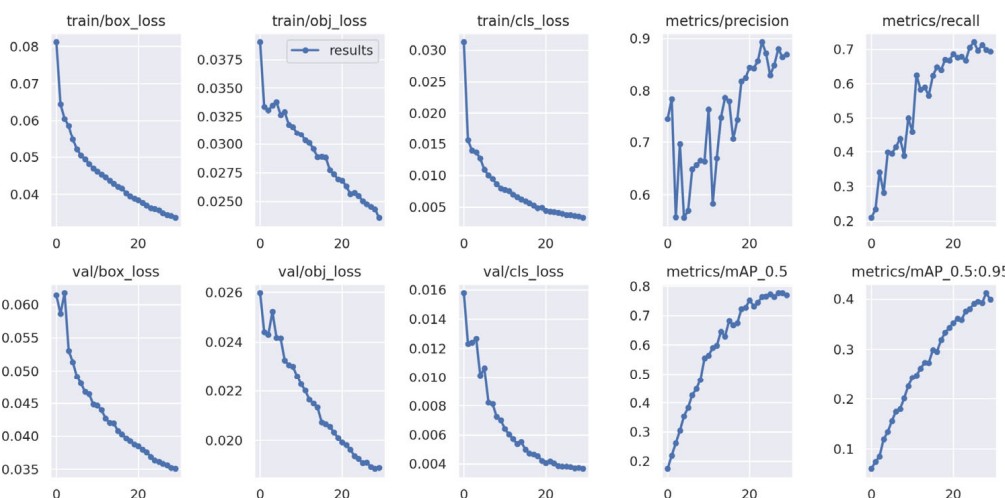

**Figure 21.** Comprehensive results of YOLOv5 model with the pretrained weights.

## 6. Conclusions

The best models of YOLOv3 with/without pretrained weights with the highest F1 score pointed out a good trade-off between precision and recall. From the results obtained, it is also evident that the YOLOv3 models, in general, are resilient to the initial weights, perform well with just 20 epochs of training, and have negligible differences in terms of performance with or without pretrained weights. In the case of YOLOv5 models, more epochs are required to obtain a model with good performance when training without any pretrained weights, which is evident from the drastic difference in performance between models trained with and without pretrained weights for 20 epochs. Thus, we can conclude that YOLOv3 architecture is suitable for applications where training resources for training a new model on a new dataset for a customized application are limited, and no pretrained weights are available. The custom dataset comprised a total of 5945 images, out of which 4482 images were used as the training set and 1463 images were used as a test set. With the quantity of images in the dataset, a high degree of accuracy was obtained. This reflects the high quality of the model proposed. The dataset used was generated firsthand, thereby creating a custom dataset which covers all the aspects needed for training and testing the model so proposed. In the future, a more extensive dataset can be generated by exploiting more resources as per feasibility to obtain a more robust model. Compared to training the dataset with the YOLOv5 architecture without any pretrained weights, YOLOv3 provides better results for both F1 Score and mAP in this particular dataset for the ADAS application. This was tested using a dataset that included various object classes detected in the video feed, including four-wheel vehicles, two-wheel vehicles, pedestrians, stray animals, road damage, unmarked speed bumps, and barricades. These models were deployed into a mobile application on the target ADAS prototype hardware. From the analysis of the various models, we can conclude that the models performed and generalized reasonably well on the collected dataset. Di Ao et al. 2022 proposed a subjective assessment for testing L2 and L2+ ADAS systems which, if used to compare the proposed system with the vision-based features of existing ADAS systems, could further provide better insights [20]. For automotive aftermarket ADAS designers and manufacturers that use computer vision-based AI algorithms to accomplish some functionality, a significant amount of their budget and time is spent on the development and testing of the model. To improve the adoption of these systems, they must be affordable, and manufacturers can do this by using transfer learning-based models (YOLO models with pretrained weights) to suit their target application and reduce the model development time with a slight trade-off in the form of accuracy. The models without pretrained weights have performed as well as their counterparts using pretrained weights, but the trade-off is increased training time. The ADAS system proposed in this module can alert the vehicle drivers of any obstacles.

The alerting mechanism adopted in this system is a buzzer along with visual feedback in the mobile application. This ADAS system can be a life saver in cases when there is a lapse in concentration from the driver's end. Such systems can help prevent road accidents on a large scale and prevent loss of life as well as vehicle damage. Car manufacturers can also think about adding the ADAS system as a feature in the infotainment systems of their cars. This ADAS feature can prove to be a valuable addition to car manufacturers.

**Author Contributions:** Conceptualization, K.J.; Data curation, C.N.J., G.P.A., A.J.S. and K.S.; Formal analysis, H.V. and K.J.; Investigation, C.N.J., A.J.S., K.S. and K.J.; Methodology, G.P.A. and A.J.S.; Project administration, K.J.; Resources, H.V. and R.K.R.; Software, H.V., K.S. and R.K.R.; Validation, P.R.; Writing—original draft, C.N.J., G.P.A., R.K.R. and K.J.; Writing—review & editing, P.R. All authors have read and agreed to the published version of the manuscript.

**Funding:** The APC is funded by Vellore Institute of Technology, Vellore-632014, India.

**Institutional Review Board Statement:** Approved by the Institute.

**Informed Consent Statement:** Not applicable for our case.

**Data Availability Statement:** Ours is a own custom dataset created by ourselves.

**Conflicts of Interest:** Authors declare no conflict of interest.

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
