# Peer review of "Implementation of Deep Learning Algorithm on a Custom Dataset for Advanced Driver Assistance Systems Applications"

_applsci, doi:10.3390/app12188927_

Round 1
Reviewer 1 Report
I am satisfied with this paper and I have a few suggestions.
1. First of all, we hope that the authors will use the template of this journal
2. Too few references in this paper. I suggest that authors cite professional intelligent vehicle-related journal papers,for example Journal of Intelligent and Connected Vehicles. Related papers, such as 'Subjective assessment for an advanced driver assistance system: a case study in China', 'Merging control strategies of connected and autonomous vehicles at freeway on-ramps: a comprehensive review', 'Longitudinal control for person-following robots'
3. The author only used 6000 images, would it be too little. Suggest the author to increase the data set.
'
Author Response
We would like to thank the reviewer for the constructive feedback:
The responses to the review comments are given below in the order of the review questions:
- We have modified our manuscript as per the Journal's format.
- We have cited the article "Subjective assessment for an advanced driver assistance system: a case study in China" in our revised manuscript as suggested and the other two articles suggested by the reviewer do not seem to be matching with our research area.
- The custom data set prepared in our work comprised a total of 5945 images, out of which, 4482 images were used as the training set and 1463 images were used as a test set. With the number of images in the data set, a high degree of accuracy was obtained. This reflects the high quality of the model proposed. The data set used was generated first-hand, thereby creating a custom data set which covers all the aspects needed for training and testing the model so proposed. In the future, a more extensive dataset can be generated by exploiting more resources as per feasibility to obtain a more robust model.
Reviewer 2 Report
The article concerns a very interesting and very important application of deep learning techniques, namely the support of automatic cars. The research results of the authors of this work are impressive, and the research techniques used are correct and convincing.
I have some comments regarding the structure of the article.
1. "Literature Survey" should be presented after "Introduction" and before "Technical Overview of Artificial Intelligence".
2. The results of other methods presented in Table 1 should be included in the chapter "Performance Comparison" at the end of the paper and should be compared directly with the results obtained in the research related to the article.
3. The overview of the proposed system (Figure 1) should be part of the "Proposed Methodology" chapter.
I believe this can improve the readability of the article. These are of course only my suggestions and the authors may not follow them.
I have no substantive comments. The structure of the neural network is accurately depicted. The network training parameters are fully described. The test application is also adequately presented. The research results are detailed and convincing. I am very impressed with the research.
Author Response
The authors would like to thank the reviewer for the constructive feedback.
The responses for the reviewer comments are as follows in the order of the review questions:
- We have shifted the Literature survey after the introduction chapter and before the Technical overview of Artificial Intelligence in the revised manuscript.
- In our original manuscript, we have compared the performance of other methods alone in the literature review chapter in Table 1 and we have also compared the performance of other methods with the results of our proposed method in the performance comparison chapter in Table 2. Now, as recommended by the reviewer, we have removed Table 1 from the Literature survey chapter and retained only Table 2 (which is now re-numbered as Table 1 in the revised manuscript) which include the results of other methods with our proposed research work and we have directly compared the performance of all the results obtained in the research related to the article in the performance comparison chapter.
- The overview of the proposed system (Figure 1) is now shifted to the Proposed Methodology chapter in the revised manuscript.